# A Hybrid Incremental Nonlinear Dynamic Inversion Control for Improving Flying Qualities of Asymmetric Store Configuration Aircraft

Chang-ho Ji [1], Chong-sup Kim [2,*] and Byoung-Soo Kim [3]

1   Flight Control Test Team, Korea Aerospace Industries, Ltd., Sacheon 52529, Gyeongsangnamdo, Korea; saerooom@koreaaero.com
2   Flight Control Law Team, Korea Aerospace Industries, Ltd., Sacheon 52529, Gyeongsangnamdo, Korea
3   School of Aerospace and Software Engineering, Gyeongsang National University, Jinju-si 52725, Gyeongsangnamdo, Korea; bskim@gnu.ac.kr
*   Correspondence: robocskim@koreaaero.com

**Abstract:** Highly maneuverability fighter aircrafts are equipped with various weapons for successful air-to-air and air-to-ground missions. The aircraft has abrupt transient response due to ejection force generated when store of one wing is launched and the movement of lateral center-of-gravity (YCG) changing by the mass distribution of both wings after launched. Under maintaining 1 g level flight with manual trim system in the asymmetric store configuration, the aircraft causes unexpected roll motion for the pure longitudinal maneuver because the change of AoA and airspeed changes the amount of trim for level flight of the aircraft. For this reason, the pilot should continuously use the roll control stick input to maintain level flight. This characteristic increases the pilot's workload and adversely affects the flying qualities of the aircraft, which is a major cause of deteriorating mission efficiency for combat maneuver. In this paper, we propose a hybrid control that combines model- and sensor-based Incremental Nonlinear Dynamic Inversion (INDI) control based mathematical model of the supersonic advanced trainer to minimize the transient response of the aircraft when the store is launched and to effectively reduce the unexpected roll motion that occurs for the pure longitudinal maneuvering in the asymmetric store configuration. As a result of the frequency- and time-domain evaluation, the proposed control method can effectively reduce the transient response for store launch and minimize unexpected roll motion for the pure longitudinal maneuver. Therefore, this control method can effectively improve flying qualities and mission efficiency by reducing the pilot's workload in the operation of the asymmetric store configuration.

**Keywords:** hybrid INDI control; flying qualities; asymmetric store configuration





## 1. Introduction

Highly maneuverable fighter aircrafts carry various stores such as missiles, bombs and fuel tanks internally or externally for air-to-air and air-to-ground missions [1]. The degradation of performance, flying qualities, structure and flutter usually occur at the store separation flight test event [2,3]. Especially, the external store installed asymmetrically on the aircraft significant affects flying qualities and flight performance by changing the weight and center of gravity of the longitudinal and lateral axis and significantly changing the basic airflow over the aircraft fuselage and control surfaces [4]. Moreover, when additional stores are carried on the aircraft, flight characteristics such as flying qualities and store separation should be verified through flight tests. Traditionally, many NATO nations use specifications such as MIL-STD-1763 [5], MIL-HBDK-1763 [6], MIL-HDBK-244A [7], NATO STANG 7068 [8] and Science and Technology (STO) AGARDograph 300 Vol 29 [9] as the basis for conducting analysis, wind tunnel testing, modelling and simulation (M&S), prior to ground and flight testing. In addition, related to flying qualities, the specification

of MIL-STD-1797A [10] recommends that the intentional release or ejection of any store shall not result in objectionable flight characteristics or impair tactical effectiveness for Level 1 and Level 2.

Most of the stores are intentionally separated by the pilot for in-flight operation [11,12]. As shown in Figure 1a, the aircraft quickly becomes asymmetric store configuration just after one store is released and the opposite store remains on the wing. For this reason, the position of the center of gravity on the lateral axis is shifted, and the difference in the lift force and mass distribution of both wings occurs. Therefore, when the store is launched asymmetrically, a sudden change in lateral-directional axis and trim appears, and the ride and flying qualities of the aircraft are degraded in proportion to the magnitude of the transient response. The flight control laws of the production fighter aircraft are designed primary for the symmetric store configuration, so the effect of lateral asymmetries represents a significant off-design condition. If the aircraft have a high lateral asymmetric configuration, the control margin may be substantially reduced. Most of the fighter aircraft maintain a 1 g level flight by setting the rolling moment to zero using manual roll trim device in asymmetric store configuration. However, the amount of roll trim required for 1 g level flight varies depending on the flight conditions such as altitude, speed and angle of attack (AoA), so the pilot should continuously use the manual roll trim system during the mission tasks. This increases the pilot's control workload and degrades the flight performance of the aircraft. Considering the flight dynamics equation [13,14], the operation of asymmetric store configurations has at least two issues in terms of flight dynamics. Firstly, the cross-coupling characteristics between the inter-axes resulting from the lateral shift in the center-of-gravity and non-zero products of inertia, $I_{xy}$ and $I_{yz}$, increase. Secondly, the asymmetric store changes aerodynamic characteristics of the aircraft such as lift of the wing. Therefore, even if the pilot maintains a 1 g level flight by using a roll trim in the specific flight condition, the influence of the lift force distribution and inertia of both wings acts differently on the aircraft as the AoA and airspeed changes due to the pure longitudinal maneuver, so the pilot should control the aircraft with the three-axis control input. For example, as shown in Figure 1b, unexpected rolling motion occurs by the pure longitudinal control input and the pilot additionally needs the roll control input to maintain the level fight. These aircraft characteristics directly affect flying qualities evaluation, such as pitch attitude capture and air-to-air tracking, and cause flying qualities degradation.

Most of the aircraft are designed to be robust to disturbances by increasing the roll rate feedback control gain to improve the flying qualities of the asymmetric store configuration. The T-50 [15,16] advanced supersonic trainer has improved flying qualities of asymmetric store configuration by designing the blend roll control system that increases the control gain of the roll rate feedback, which is in a small range of roll control input and roll rate of the aircraft, to reduce the roll control input required for pure longitudinal axis maneuvering. The F-18 E/F Hornet [17] adopted rolling surface-to-rudder interconnect (RSRI) for control to reduce the cross-coupling characteristics between axes. Moreover, the EF-2000 [18] and JAS-39 [19] improved flying qualities to reduce the pilot's maneuvering workload in the asymmetrical store configuration by designing the automatic trim function in all axes. These control methods should be premised on securing an accurate mathematical model of the aircraft and a limited method on reducing the transition response for store launch and the rolling motion resulting from the change of AoA and airspeed in the pure longitudinal maneuver.

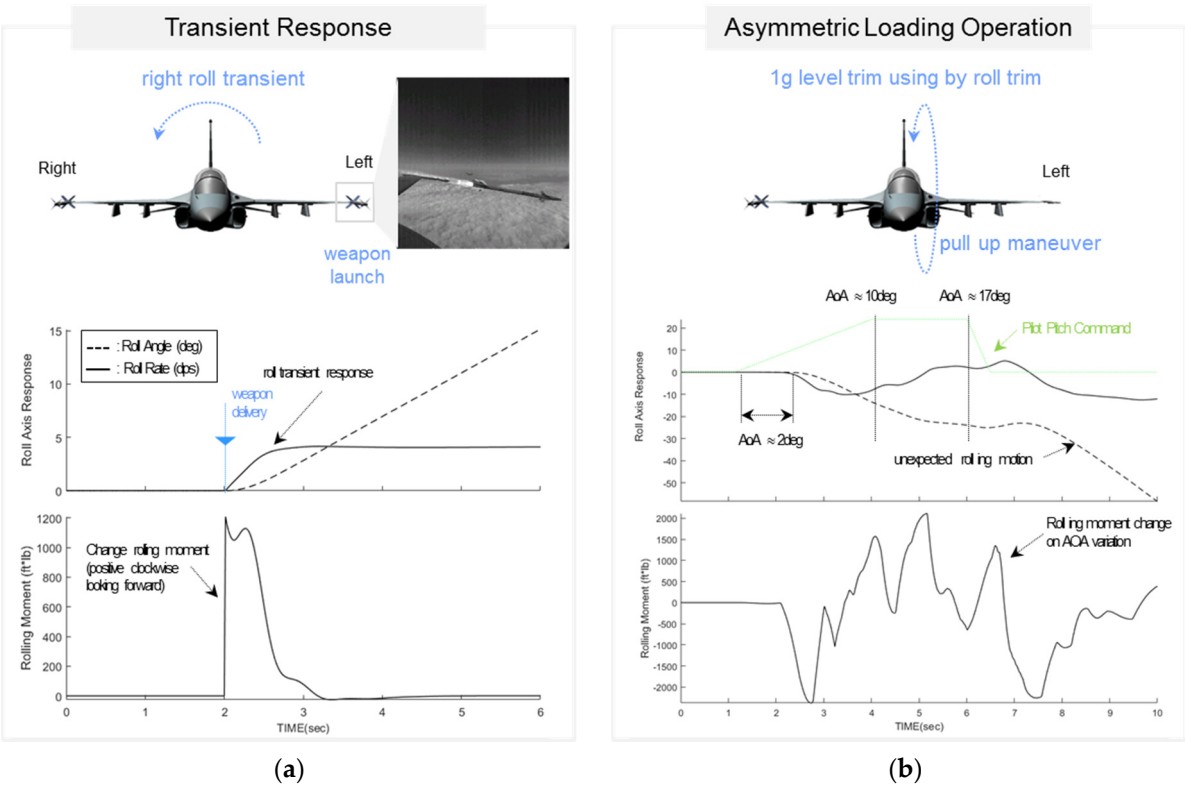

**Figure 1.** Definition of unexpected response for weapon launches and asymmetric store configurations (**a**) transient due to weapon launch, (**b**) unexpected rolling motion for pull up maneuver of asymmetric store configuration.

To mitigate abrupt transient response to asymmetric store launch and unexpected rolling motions for pure longitudinal maneuver in asymmetric store configuration, we propose the hybrid Incremental Nonlinear Dynamic Inversion (INDI) control, which combines a model- and sensor-based INDI control. The main contribution of this control method proposed in this paper can be summarized as follows: Firstly, the control methods effectively reduce roll transient response in case of store launch of the one wing. Therefore, ride qualities of the aircraft are improved and decrease the magnitude of roll trim to maintain 1 g wing level flight after store launch. Secondly, it is possible to effectively reduce the unexpected roll response that occurs during longitudinal axis maneuvering, thereby reducing the pilot's control workload and improving flying qualities of the aircraft. Thirdly, this control method is fairly robust against uncertainties of major aerodynamic properties, compared to other classical control methods. Lastly, the hybrid INDI control is fundamentally based on the proven model-based INDI, so that it can easily be applied to production aircraft to ensure flight safety and further improve control performance of the aircraft.

The rest of this paper is organized as follows. Sections 2–6 describes the control theory of the Hybrid INDI control methods and the result of designing the synchronization filter of the control surface feedback path. Section 7 describes the evaluation flight conditions and methods and shows the evaluation results of the proposed control methods with the frequency-domain stability assessment and the time-domain nonlinear simulation results based on the mathematical model of the advanced supersonic trainer. In addition, Section 8 presents conclusions and future plans.

## 2. Control Law Design

Recently, INDI has been extensively studied and applied to demonstration and production of aircraft. The development of a model-based NDI control in aerospace industry started in National Aeronautics and Space Administration (NASA) with the participation

of Honeywell, Boeing, and Lockheed Martin in early 1990s. The use of model-based NDI as a viable control law methodology has been demonstrated in the restricted flight envelope on various flight control research aircraft such as F-18 high angle-of-attack research vehicle (HARV) [20], X-38 [21,22], X-36 reconfigurable control for tailless aircraft (RESTORE) [23] and X-35B short take-off/vertical landing (STOVL) [24]. In addition, F-35 Joint Strike Fighter (JSF) [25] was the first production fighter incorporating model-based INDI in entire flight envelopes. Moreover, sensor-based INDI which uses the measured angular acceleration and control surface positions as feedback parameters was evaluated on vectored thrust aircraft advanced control (VAAC) Harrier [26,27] in 1999. In 2000, NASA applied this control method to innovative control effector tailless aircraft [28]. Recently, the Netherlands Aerospace Centre (NLR) and German Aerospace Center (DLR) with the Technical University of Delft have applied it to Cessna 550 demonstrator [29] and proved the performance of the developed control law. The stability and robustness of the sensor-based INDI control has already been proven [30,31].

Figure 2 shows the control structure of the hybrid INDI control that combines a model- and sensor-based INDI control. The model-based INDI represents decoupling of flying quality-dependent portions from the design of the one dependent on airframe dynamics. The desired angular acceleration ($\dot{x}_{des}$) is calculated from desired dynamics which reflects how the aircraft should fly in response to the pilot input. The desired dynamics consist of command shaping and regulator. The command shaping aims to translate the pilot stick input to the desired aircraft movement, while the regulator aims to directly set the Low-Order Equivalent System (LOES) parameter values such as roll mode time constant, Dutch-roll mode damping and natural frequency to comply with the classical flying qualities criteria while the aircraft is achieving this motion. The airframe-dependent portion comprises On-Board Model (OBM) and Control Allocation (CA). The OBM provides the estimated angular acceleration to calculate the dynamic inversion and control effectiveness matrix to compute CA.

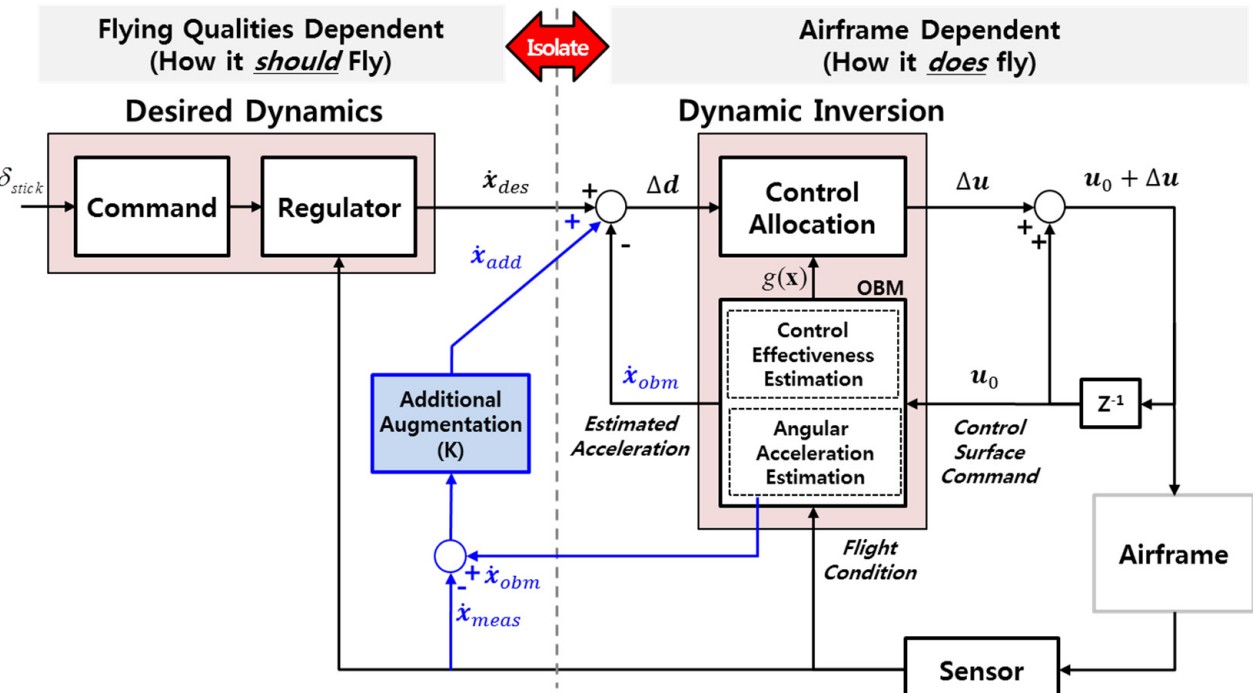

**Figure 2.** Control structure of the hybrid INDI control.

The nonlinear dynamic equation of motion can be expressed as

$$\dot{x} = F(\boldsymbol{x}, \boldsymbol{u}) \tag{1}$$

where $x \in R^n$ is the state vector, and $u \in R^m$ is the control input vector. In general, the state vector includes the angular velocity of the aircraft. Equation (1) can be rewritten as

$$\dot{x} = f(x) + g(x)u \tag{2}$$

where $f$ is a nonlinear state dynamic function and $g$ is a nonlinear control distribution function. If actual control command $u$ is defined as the sum of previous control command $u_0$ and incremental control command $\Delta u$, Equation (2) can be shown as Equation (3).

$$\dot{x} = f(x) + g(x)(u_0 + \Delta u) \tag{3}$$

If we assume $g(x)$ is invertible for all values of $x$, Equation (3) can be summarized as Equation (4)

$$\Delta u = g_{obm}^{-1}(x)\left\{ \dot{x} - (f_{obm}(x) + g_{obm}(x)u_0) \right\} \tag{4}$$

where $f_{obm}(x) + g_{obm}(x)u_0$ is angular acceleration $\dot{x}_{obm}$ calculated from OBM. We will specify $\dot{x}$ as the rate of the desired states $\dot{x}_{des}$ to achieve the flying qualities design goals defined by designer. By swapping $\dot{x}$ in the previous equation to $\dot{x}_{des}$, Equation (4) can be arranged as Equation (5).

$$\Delta u = g_{obm}^{-1}(x)\left\{ \dot{x}_{des} - \dot{x}_{obm} \right\} \tag{5}$$

Consequently, the current control command $u_{cmd}$ can be designed by combining the previous control command and the increment control command, as shown in Equation (6)

$$u_{cmd} = u_0 + \Delta u \tag{6}$$

By substituting Equation (4) into Equation (3), the dynamic characteristics of the aircraft can be completely canceled, and the desired angular acceleration of the aircraft can be obtained as

$$\dot{x} = f(x) + g(x)\left\{ u_0 + g_{obm}^{-1}(x)\left( \dot{x}_{des} - \dot{x}_{obm} \right) \right\} = \dot{x}_{des} \tag{7}$$

Note also that if the exact aircraft model can be obtained, the desired dynamics that depend on the flying qualities requirements can also be designed without considering the aircraft dynamics. However, it is impossible to obtain the exact aircraft model due to several models such as computational time delay and actuator and sensor dynamics in control system. As these factors increase the order of control system due to the complex models, they also increase the errors between the aircraft model and the flying qualities requirements represented in first- or second-order models. This means that the flying quality parameters in desired dynamics must be adjusted based on the off-line optimization procedure to compensate for uncertainties of aircraft dynamics.

## 3. Desired Dynamics

Figure 3 shows the detailed structure of the hybrid INDI control in lateral-directional axis. The desired dynamics are designed on a basis of proportional control with stability-axis roll rate $p_s(°/s)$, sideslip $\beta(°)$ and sideslip rate sideslip $\dot{\beta}(°/s)$ as feedback variables. Figure 3b shows the control architecture of desired dynamics for lateral-directional axis. The response types are selected as stability-axis roll rate to achieve fast roll response in lateral axis and sideslip to augment Dutch-roll damping and frequency in directional axis [29].

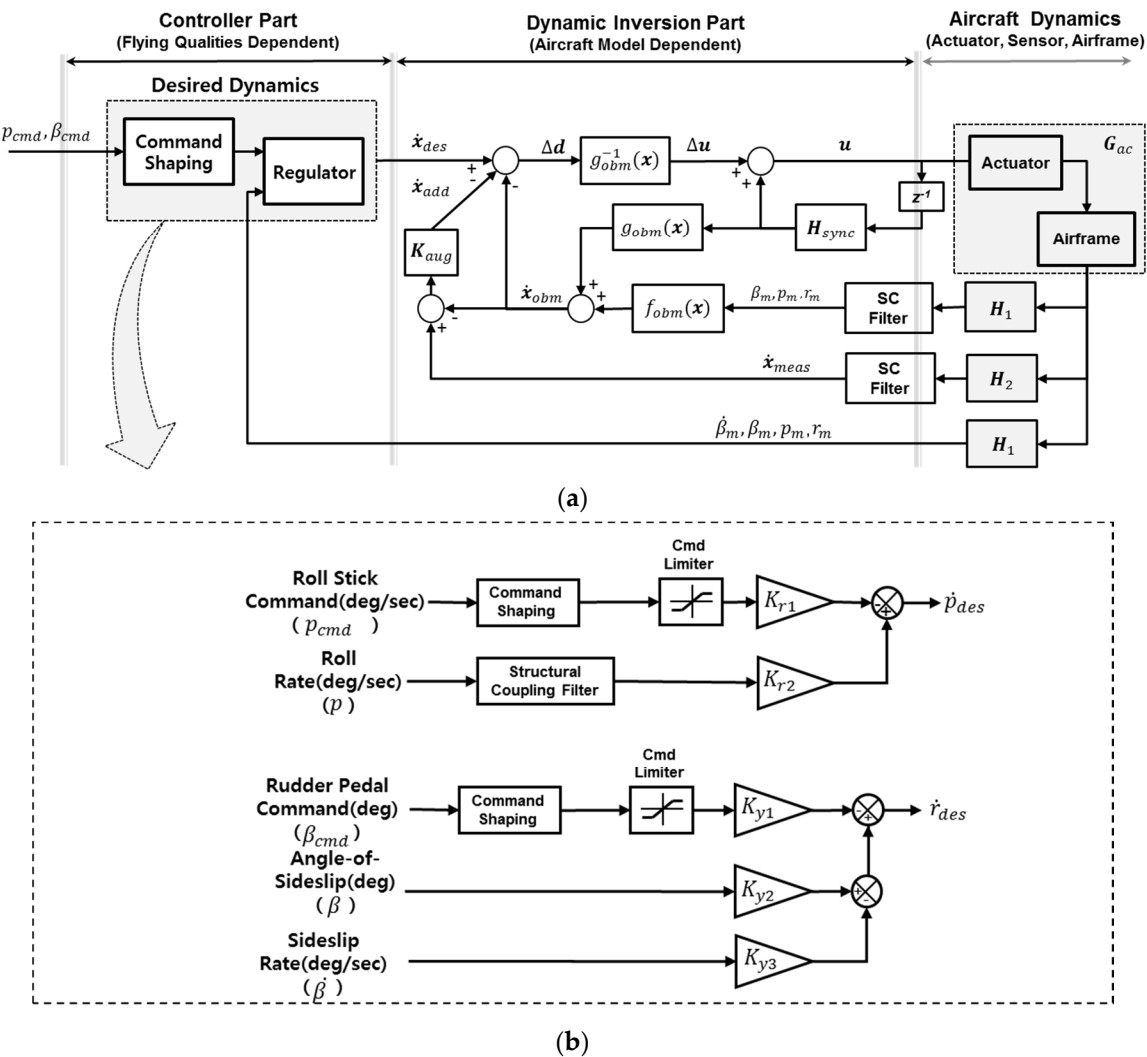

**Figure 3.** Detailed structure of the Hybrid INDI control in lateral-directional axis (**a**) dynamic inversion part (**b**) desired dynamics part.

The initial value of flying quality parameters ($K_{r1}$, $K_{r2}$, $K_{y1}$, $K_{y2}$ and $K_{y3}$) can be obtained as

$$K_{r1} = K_{r2} \frac{p_{s,max}}{p_{s,cmd,max}}, \quad K_{r2} = -\tau_{roll}, \tag{8}$$

$$K_{y1} = K_{y2} \frac{\beta_{max}}{\beta_{cmd,max}}, \quad K_{y2} = \omega_{dr}, \quad K_{y3} = -2\zeta_{dr}\omega_{dr} \tag{9}$$

where $p_{s,max}$ and $p_{s,cmd,max}$ are the maximum roll rate and roll rate command, $\beta_{max}$ and $\beta_{cmd,max}$ are the maximum sideslip and sideslip command. $\tau_{roll}$ is roll time constant and $\zeta_{dr}$, and $\omega_{dr}$ are the dutch-roll mode damping and natural frequency as a design goals of lateral directional control law.

## 4. Inverse Model

In general, lateral-directional equations of motion can be expressed as Equations (10) and (11).

$$\dot{p} = \frac{I_{zz}L + I_{xz}N}{I_{xx}I_{zz} - I_{xz}^2} + \frac{I_{xz}(I_{xx} - I_{yy} + I_{zz})pq}{I_{xx}I_{zz} - I_{xz}^2} + \frac{[I_{zz}(I_{yy} - I_{zz}) - I_{xz}^2]qr}{I_{xx}I_{zz} - I_{xz}^2} \tag{10}$$

$$\dot{r} = \frac{I_{xz}L + I_{xz}N}{I_{xx}I_{zz} - I_{xz}^2} + \frac{I_{xz}(I_{xx} - I_{yy} + I_{zz})qr}{I_{xx}I_{zz} - I_{xz}^2} + \frac{[I_{xx}(I_{xx} - I_{yy}) - I_{xz}^2]pq}{I_{xx}I_{zz} - I_{xz}^2} \tag{11}$$

where $p$, $q$ and $r$ $(°/s^2)$ are the angular velocities of the aircraft; $I_{xx}$, $I_{yy}$ and $I_{zz}$ are the principal moments of inertia, and $I_{xz}$ is the product of inertia. Now, we will assume the lateral-directional moments $L$ and $N$ are linear with respect to aerodynamic derivatives, i.e.,

$$L = L_\beta\beta + L_p p + L_r r + L_{\delta_{aa}}\delta_{aa} + L_{\delta_r}\delta_r + L_{\delta_{ea}}\delta_{ea} \tag{12}$$

$$N = N_\beta\beta + N_p p + N_r r + N_{\delta_{aa}}\delta_{aa} + N_{\delta_r}\delta_r + N_{\delta_{ea}}\delta_{ea} \tag{13}$$

where $\delta_{ea}$, $\delta_{aa}$ and $\delta_r$ are the asymmetric deflections of the horizontal tails (HT), trailing edge flaps (TEF) and rudder. By substituting the above linear moment equations into Equations (10) and (11), we can obtain a relation in Equation (14) that combines linear and nonlinear terms.

$$\begin{bmatrix} L'_{\delta_{ea}} & L'_{\delta_{aa}} & L'_{\delta_r} \\ N'_{\delta_{ea}} & N'_{\delta_{aa}} & N'_{\delta_r} \end{bmatrix} \begin{bmatrix} \delta_{ea} \\ \delta_{aa} \\ \delta_r \end{bmatrix} = \left\{ \begin{bmatrix} \dot{p} \\ \dot{r} \end{bmatrix} - \begin{bmatrix} L'_\beta & L'_p & L'_r \\ N'_\beta & N'_p & N'_r \end{bmatrix} \begin{bmatrix} \beta_0 \\ p_0 \\ r_0 \end{bmatrix} - \begin{bmatrix} I_{xx} & I_{xz} \\ -I_{xz} & I_{zz} \end{bmatrix}^{-1} \begin{bmatrix} I_{xz}p_0q_0 + (I_{yy} - I_{zz})q_0r_0 \\ I_{xz}q_0r_0 + (I_{xx} - I_{yy})p_0q_0 \end{bmatrix} \right\} \tag{14}$$

where $L'_k$ and $N'_k$ are linearized moments due to aerodynamic forces and can be defined as

$$L'_k = \frac{I_z\left(\frac{\partial}{\partial k}\right)L + I_{xz}\left(\frac{\partial}{\partial k}\right)N}{I_{xx}I_{zz} - I_{xz}^2}, \quad k = \beta, p, r, \delta_{ea}, \delta_r \tag{15}$$

$$N'_k = \frac{I_{xz}\left(\frac{\partial}{\partial k}\right)L + I_z\left(\frac{\partial}{\partial k}\right)N}{I_{xx}I_{zz} - I_{xz}^2}, \quad k = \beta, p, r, \delta_{ea}, \delta_r \tag{16}$$

If the last term is ignorable in Equation (14), the result is identical to the linear set of the DI equations. Finally, inverting the above equation as well as applying the commanded, desired, and measured values gives the resulting the INDI control.

$$\begin{bmatrix} \delta_{ea} \\ \delta_{aa} \\ \delta_r \end{bmatrix}_{cmd} = \begin{bmatrix} L'_{\delta_{ea}} & L'_{\delta_{aa}} & L'_{\delta_r} \\ N'_{\delta_{ea}} & N'_{\delta_{aa}} & N'_{\delta_r} \end{bmatrix}^{-1}$$
$$\left\{ \begin{bmatrix} \dot{p} \\ \dot{r} \end{bmatrix}_{des} - \begin{bmatrix} L'_\beta & L'_p & L'_r \\ N'_\beta & N'_p & N'_r \end{bmatrix} \begin{bmatrix} \beta_0 \\ p_0 \\ r_0 \end{bmatrix}_{meas} - \begin{bmatrix} I_{xx} & I_{xz} \\ -I_{xz} & I_{zz} \end{bmatrix}^{-1} \begin{bmatrix} I_{xz}p_{0meas}q_{0meas} + (I_{yy} - I_{zz})q_{0meas}r_{0meas} \\ I_{xz}q_{0meas}r_{0meas} + (I_{xx} - I_{yy})p_{0meas}q_{0meas} \end{bmatrix} \right\} \tag{17}$$

## 5. Additional Augmentation

In the control structure of Figure 3a, the angular acceleration calculated from the OBM ($\dot{x}_{obm}$), the angular acceleration obtained from additional augmentation control ($\dot{x}_{add}$) and the virtual control command ($\Delta d$) are given as

$$\dot{x}_{obm} = f_{obm}(x) + g_{obm}(x)u_0 \tag{18}$$

$$\dot{x}_{add} = K_{aug}(\dot{x}_{meas} - \dot{x}_{obm}) \tag{19}$$

$$\Delta d = \dot{x}_{des} - \dot{x}_{obm} - \dot{x}_{add} \tag{20}$$

where $K_{aug}$ is an n-dimensional diagonal matrix, which means the control gain of additional augmentation control. Each element of $k_i$ has an arbitrary value between 0.0 and 1.0. By substituting from Equations (18)–(20) into Equation (6), the current control command can be obtained as

$$u = u_0 + g_{obm}^{-1}(x)\left[\dot{x}_{des} - \left\{K_{aug}\dot{x}_{meas} + (I - K_{aug})\dot{x}_{obm}\right\}\right] \tag{21}$$

where the term of $K_{aug}\dot{x}_{meas} + (I - K_{aug})\dot{x}_{obm}$ means to use $\dot{x}_{meas}$ and $\dot{x}_{obm}$ proportionally according to the value of $K_{aug}$. By substituting from Equation (21) into Equation (2), the dynamic equation of motion including control law can be expressed as

$$\dot{x} = f(x) + g(x)u_0 + \left[\dot{x}_{des} - \left\{K_{aug}\dot{x}_{meas} + (I - K_{aug})\dot{x}_{obm}\right\}\right] \tag{22}$$

Generally, the hybrid INDI control is a control synthesis technique in which the inherent dynamics of a dynamical system cancel out and replace the desired dynamics, selected by control law designer. However, the plant dynamics cannot be modeled exactly in real world, thereby preventing an exact replacement of the inherent plant dynamics with the desired dynamics. For any $1 \leq i \leq n$ and $1 \leq j \leq m$

$$min\{e_{i,meas}, e_{i,obm}\} \leq f_i(x) + \sum_{j=1}^{m} g_{ij}(x)u_0 - \left[\{k_i\dot{x}_{i,meas} + (1 - k_i)\dot{x}_{i,obm}\}\right] \leq max\{e_{i,meas}, e_{i,obm}\} \tag{23}$$

where $f_i(x)$ is the $i$-th element of $f(x)$, and $g_{ij}(x)$ is the $(i,j)$-th element of $g(x)$. In addition, $k_i \in \{0,1\}$ is the $(i,j)$-th element of $k$. Moreover, $\dot{x}_{i,meas}$ and $\dot{x}_{i,obm}$ are the $i$-th element of $\dot{x}_{meas}$ and $\dot{x}_{obm}$, respectively. The error of the aircraft model is given by

$$e_{i,meas} = f_i(x) + \sum_{j=1}^{m} g_{ij}(x)u_0 - \dot{x}_{i,meas} \tag{24}$$

$$e_{i,obm} = f_i(x) + \sum_{j=1}^{m} g_{ij}(x)u_0 - \dot{x}_{i,obm} \tag{25}$$

The additional augmentation control can apply angular acceleration error to control law, which is always smaller than the maximum error between the angular acceleration calculated from the OBM or measured from sensor and the angular acceleration of the actual aircraft. That is, the ratio of $\dot{x}_{obm}$ is increased in the subsonic and supersonic flight regions where accurate model estimation is available, and the ratio of $\dot{x}_{meas}$ is increased in high angle-of-attack and transonic flight regions where it is very difficult to estimate an accurate model in unsteady flow fields. Therefore, this prevents the deterioration of flying qualities due to the maximum angular acceleration error applied to the controller.

## 6. Control Surface Synchronization Filter

The inertial measurement unit (IMU) sensor is never infinitely fast, which degrades performance and necessitates the use of synchronization filter; this section describes the synchronization filter design method. In the hybrid INDI control approach, the feedback for angular acceleration is assumed to be a linear combination of the measured angular acceleration from IMU sensor measurements and the estimated angular acceleration from OBM. The complete transfer function from the desired control input to the elevator control command is given by

$$\frac{u}{\dot{x}_{des}} = \frac{G_{ac}^u}{\left\{\dot{x}_{fb} - g_{obm}(x)H_{sync}u\right\}e^{-\delta T} + g_{obm}(x)} \tag{26}$$

where $H_{syn}$ is 3 by 3 diagonal matrix, and order of elements are 4th order synchronization filter. The total angular acceleration feedback $\dot{x}_{fb}$ can be expressed as

$$\begin{aligned}
\dot{x}_{fb} &= K_{aug}\dot{x}_{meas} + (I - K_{aug})\dot{x}_{obm} \\
&= K_{aug}\left\{f_{meas}(x)H_2^{total}x + g_{meas}(x)H_2^{total}\right\} \\
&\quad + (I - K_{aug})\left\{f_{obm}(x)H_1^{total}x + g_{obm}(x)H_{sync}u\right\}
\end{aligned} \tag{27}$$

where $H_1^{total}$ are sideslip and rate sensor models, and $H_2^{total}$ is angular acceleration sensor model including aircraft and actuator dynamics. Considering the denominator of Equation (26) by substituting from Equation (27) into Equation (26),

$$D_{\dot{q}} = \left\{ \boldsymbol{K_{aug}} f_{meas}(\boldsymbol{x}) \boldsymbol{H_2^{total}} + (\boldsymbol{I} - \boldsymbol{K_{aug}}) f_{obm}(\boldsymbol{x}) \boldsymbol{H_1^{total}} \right\} \boldsymbol{x} e^{-\delta T}$$
$$+ \left\{ \boldsymbol{K_{aug}} g_{meas}(\boldsymbol{x}) \boldsymbol{H_2^{total}} + (\boldsymbol{I} - \boldsymbol{K_{aug}}) g_{obm}(\boldsymbol{x}) \boldsymbol{H_{sync}} - g_{obm}(\boldsymbol{x}) \boldsymbol{H_{sync}} \right\} \boldsymbol{u} e^{-\delta T} + g_{obm}(\boldsymbol{x}) \tag{28}$$

In particular, the following terms require special attention

$$\boldsymbol{\Gamma} = \boldsymbol{K_{aug}} g_{meas}(\boldsymbol{x}) \boldsymbol{H_2^{total}} + (\boldsymbol{I} - \boldsymbol{K_{aug}}) g_{obm}(\boldsymbol{x}) \boldsymbol{H_{sync}} - g_{obm}(\boldsymbol{x}) \boldsymbol{H_{sync}} \tag{29}$$

These additional dynamics, which arise uniquely for sensor-based INDI, have a direct impact on the broken-loop inversion response and play an important role in the distortion of the stability margins of the complete control law.

$$\boldsymbol{H_{syn}} = \frac{\boldsymbol{K_{aug}} g_{meas}(\boldsymbol{x})}{g_{obm}(\boldsymbol{x}) - (\boldsymbol{I} - \boldsymbol{K_{aug}}) g_{obm}(\boldsymbol{x})} \boldsymbol{H_2^{total}} \quad (\boldsymbol{\Gamma} = 0;) \tag{30}$$

Assuming an ideal on-board representation of the control effectiveness ($g_{obm}(\boldsymbol{x}) \approx g_{meas}(\boldsymbol{x})$), these effects can be eliminated by carefully matching, i.e., synchronizing, the relative phase lag and time delay between the angular acceleration and actuator feedback signals.

Note that in case of $g_{obm}(\boldsymbol{x}) \neq g_{meas}(\boldsymbol{x})$, $\Gamma$ will be nonzero even in case of perfect synchronization. This equation implies that inversion loop distortion effects can only be prevented in case all high-order dynamics represented by actuator model, sensor model, differentiation filter and SCF are taken into account. However, this matching requirement comes at the cost of additional computational complexity. In case computational complexity forms a significant limitation of the control system, alternative, low-order synchronization filter designs can be adopted. However, the fact that $\Gamma \neq 0$ implies that stability margin distortions is inherent to this kind of solutions. The configuration selects a 4th low-pass filter with $\zeta_{syn} = 0.707$ for synchronization purposes when the control surface command is fed back before the actuator dynamics.

$$\boldsymbol{H_{syn}} = \left( \frac{\omega_{syn}^2}{s^2 + 2\zeta_{syn}\omega_{syn}s + \omega_{syn}^2} \right) \left( \frac{\omega_{syn}^2}{s^2 + 2\zeta_{syn}\omega_{syn}s + \omega_{syn}^2} \right) \tag{31}$$

In this study, it is assumed that the angular acceleration is obtained by differentiating the angular rate measured by IMU sensor, which is usually used to production aircraft. Here, time constant of the 1st order differential filter is selected as 0.047 (3/64) considering the characteristics of noise amplification and 64 Hz sampling time of FLCC. The second-order SCF is designed on feedback path of angular acceleration to eliminate the high-frequency structural coupling effect. As mentioned earlier, the asynchronization between the measured angular acceleration and the control surface feedbacks reduces stability margin of the control system and causes aggravating instability of the aircraft. For synchronizing of control surfaces feedback to eliminate time delay of measured angular acceleration feedback, we consider 4th order equivalent synchronization filters with a damping ratio of 0.707 on the feedback path of control surface. At this time, the frequency of the synchronization filters was optimized in two ways as follows: The first method is to optimize the frequency of synchronization filter in the dynamics of angular acceleration feedback path considering IMU sensor model to measuring angular rate, differential and SCF. Another method is to optimize the frequency of the synchronization filter so that the stability margin criteria (gain margin $\geq \pm 6$ db, phase margin $\geq \pm 45°$) can be satisfied in the linear control system environment including control law, actuator, airframe and sensor dynamics [32].

Figure 4 shows the gain and phase responses of the high-order and low-order equivalent synchronization filter at the control surface command feedback path before actuator dynamics. The green dotted line shows the phase response of 2nd-order equivalent synchronization filter matching the one of high-order system including actuator dynamics. The frequency of synchronization filter is 14.7 rad/s, but mismatch cost value is very high

as 160. To improve the phase response fitting to reduce mismatch cost value, we considered 4th order synchronization filter instead of 2nd order one. The solid blue line and the dotted red line show the phase response of 4th order synchronization filter. The frequency of 4th-order synchronization filter matching the phase response of the high-order system is 22.99 rad/s, and mismatch cost value is less than 30, which is relatively well matched below 10 rad/s frequency band.

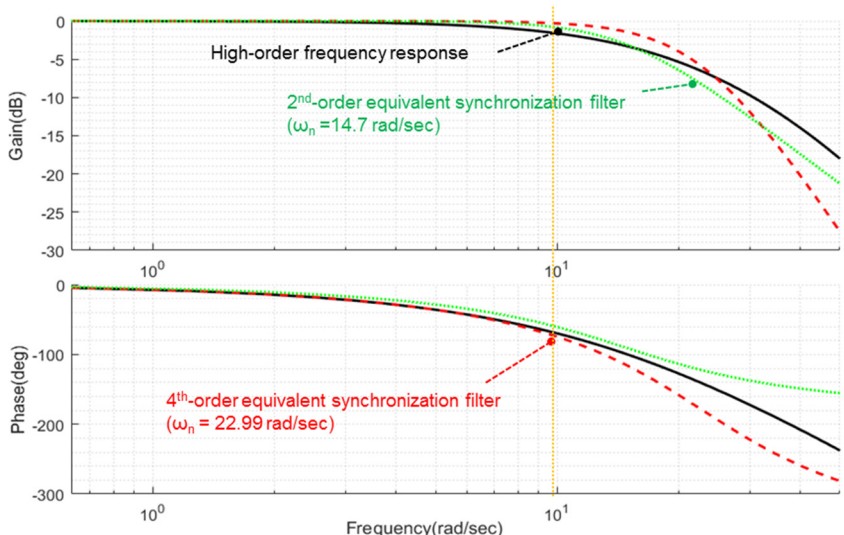

**Figure 4.** Bode plot on high-order frequency response and 4th-order equivalent synchronization filter when the control surface command is fed back before the actuator dynamics.

## 7. Analysis and Evaluation Result

### 7.1. Test Aircraft Configuration and Flying Quality Parameters

Figure 5 shows the store configurations of the supersonic fighter aircraft. This aircraft has several store configurations to support combat missions and training student pilots. The symmetric and asymmetric store configurations are representative forms of category I. The aircraft has AIM-9's at wing-tips (station 1, 7) in the symmetric store configuration and a launcher at station 1 and an AIM-9 at station 7 in the asymmetric store configuration.

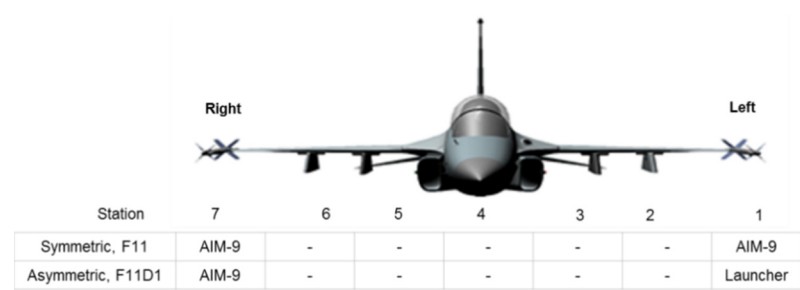

| Station | 7 | 6 | 5 | 4 | 3 | 2 | 1 |
|---|---|---|---|---|---|---|---|
| Symmetric, F11 | AIM-9 | - | - | - | - | - | AIM-9 |
| Asymmetric, F11D1 | AIM-9 | - | - | - | - | - | Launcher |

**Figure 5.** Test aircraft configuration.

The various analyzes are performed under Mach number 0.8, altitude 20 Kft and 1 g level flight condition with the symmetric and asymmetric store configurations in order to evaluate the transient response and unexpected roll motion for stores launches and longitudinal axis maneuver. The frequency-domain linear analysis is performed to assess the stability margin as well as the flying qualities based on LOES criteria, such as dutch-roll frequency and damping, roll time constant, spiral root, roll time delay, and angle-of-sideslip time delay, on the control system with the $K_{aug}$ variation and synchronization filter. Moreover, sensitivity analysis on various aerodynamic and control effectiveness uncertainties

is performed to evaluate the robustness of the control system. In addition, the time-domain nonlinear 6-DOF simulation is performed to evaluate the transient response when the store is launched in the left side wing of the symmetric store configuration and the unexpected rolling motion caused by the longitudinal axis maneuvering in asymmetric store configuration.

Table 1 shows the optimized control gains of the lateral-directional control law and the amount of trim to maintain 1 g level flight for the aircraft with the asymmetric store configuration at Mach number 0.8 and altitude 20 Kft and 1 g level flight. The displacement of asymmetric control surfaces required to keep the 1 g level flight is 0.37° for the asymmetric HT, −0.03° for the asymmetric TEF and 0.59° for the rudder. Therefore, the amount of roll and yaw trims are −3.52°/s and −0.54° to maintain 1 g level flight without control stick when the $K_{aug}$ is 0.0.

**Table 1.** Optimized flying quality parameters and trim data to 1-g wing level flight in asymmetric loading configuration (M0.8, 20 Kft altitude, 1 g level flight, asymmetric store configuration).

| $K_{aug}$ | Control Surface Deflection (°) | | | Flying Quality Parameters | | | | | Trim | |
|---|---|---|---|---|---|---|---|---|---|---|
| | Asym. TEF | Asym. HT | Rud | Kr1 | Kr2 | Ky1 | Ky2 | Ky3 | Roll (°/s) | Yaw (°) |
| 0.0 | | | | | | | | | −3.52 | −0.54 |
| 0.2 | | | | | | | | | −2.81 | −0.49 |
| 0.4 | −0.03 | 0.37 | 0.59 | −3.3 | −3.3 | −18.1 | 18.8 | −8.0 | −2.11 | −0.44 |
| 0.6 | | | | | | | | | −1.41 | −0.39 |
| 0.8 | | | | | | | | | −0.70 | −0.34 |
| 1.0 | | | | | | | | | 0.00 | 0.00 |

However, the amount of trims decreases linearly as the $K_{aug}$ value increases and become zero when the $K_{aug}$ is 1.0. In other words, there is no need for a manual trim control system since the aircraft automatically can maintain 1 g level flight.

In asymmetric store configuration, the amount of trims to maintain 1 g level flight depends on flight conditions, such as airspeed, AoA, etc. Figure 6 shows the amount of roll and yaw trims according to airspeed and AoA to maintain a 1 g level flight at an altitude of 20 Kft in the asymmetrical store configuration. At the AoA of 1.6° and an airspeed of 373 knots, the amount of roll and yaw trims for 1 g level flight are −3.52°/s and −0.54°, respectively. At the speed reduced to 300 knots and the AoA increased by 3.0°, the roll and yaw trims decreased to 0.8°/s and −0.3°. However, the roll trim increases in the positive (+) direction, and the yaw trim decreases in the negative (-) direction as the speed decreases to 199 knots and the AoA increases to 7.5° while the roll trim decreases as the speed decreases to 136 knots and the AoA increases to 17.5°. Considering these results, the pilot should continuously use the roll control stick command according to changes in airspeed and the AoA in order to maintain 1 g level flight for pure longitudinal maneuvering in asymmetric store configuration, which increases the pilot's workload and reduces mission efficiency for the combat mission.

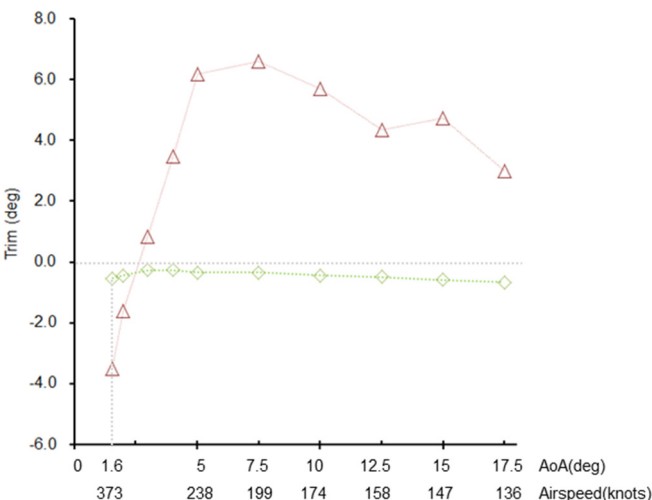

**Figure 6.** Required trim as a variation of airspeed and AoA to 1 g level flight in 20 Kft altitude in the asymmetric store configuration.

### 7.2. LOES (Low-Order Equivalent System) Analysis

Specifications such as Dutch-roll damping/frequency, roll time constant, spiral root and equivalent time delay evaluate major aircraft mode analysis based on the low-order equivalent system (LOES) analysis and are present in MIL-STD-1797A [10]. They are evaluated at each iteration of the control law optimization and are guaranteed to be met for an optimized design. The levels of flying qualities can be divided into level 1, 2 and 3, where level 1 of flying qualities means the compliance with the design goals with satisfactory flying qualities. This section describes the results of evaluating the frequency-domain analysis as a variation of the $K_{aug}$ and the effect of the synchronization filter on the control surface feedback in Mach number 0.8 and altitude 20 Kft, 1 g level flight condition, representatively. Here, the $K_{aug}$ of 0.0 means that the angular acceleration measured by IMU sensor is not used at all, and the $K_{aug}$ of 0.6 means that the measured angular acceleration is used at 60%.

Table 2 shows the results of the LOES analysis for Dutch-roll frequency and damping, roll time constant, spiral root and equivalent time delay as a variation of $K_{aug}$ and the effect of the synchronization filter on the control surface feedback. The result of the LOES analysis can be summarized as follows.

**Table 2.** Result of equivalent system analysis as a variation of additional augmentation gain and control surface synchronization in Mach 0.8, altitude 20 K, 1 g level flight.

| $K_{aug}$ | Surface Synchro | Dutch Roll | | Roll Time Constant (s) | Spiral Root (s) | Roll Time Delay (s) | AoS Time Delay (s) | Cost | HQ |
| --- | --- | --- | --- | --- | --- | --- | --- | --- | --- |
| | | Frequency (rad/s) | Damping | | | | | | |
| 0.0 | on | 10.12 | 0.78 | 0.37 | 0.00 | 0.07 | 0.07 | 2.54 | Level 1 |
| 0.6 | on | 12.55 | 0.67 | 0.37 | 0.00 | 0.05 | 0.04 | 6.47 | Level 1 |
| 0.6 | off | 4.62 | 1.61 | 0.70 | 0.00 | 0.10 | 0.13 | 16.73 | Level 2 |
| 0.8 | on | 12.21 | 0.64 | 0.38 | 0.00 | 0.03 | 0.04 | 8.51 | Level 1 |
| 1.0 | on | 10.01 | 0.71 | 0.40 | 0.00 | 0.03 | 0.08 | 39.27 | Level 1 |

Firstly, the flying qualities is improved when the $K_{aug}$ is set to 0.6 compared to when the $K_{aug}$ is set to 0.0, where 4th-order synchronization filter with a damping ratio of 0.707 and a natural frequency of 22.9 rad/s is applied in the control surface feedback path. In particular, the equivalent time delay is significantly reduced to 0.05 s for the roll and 0.04 s for the angle of sideslip (AoS), while the Dutch-roll mode and roll time constant

characteristics are similar to those of the case when the $K_{aug}$ is set to 0.0. Secondly, the hybrid INDI control with the $K_{aug}$ of 0.6 not satisfies level 1 flying qualities criterion in case that synchronization filter is not applied to the control surface feedback path. In this case, the Dutch-roll frequency decreases from 12.55 rad/s to 4.62 rad/s, and the damping ratio increases from 0.67 to 1.61 compared to when synchronization filter is applied. In addition, the roll performance of the aircraft decreases as the roll time constant increases from 0.37°/s to 0.7°/s, the equivalent time delay of the control system increases significantly from 0.04 s to 0.10 s for the roll and from 0.04 s to 0.13 s for the AoS. Considering these results, it is needed to synchronize between the control surface feedback and the angular acceleration feedback measured from the IMU. Lastly, the mismatch cost value, which is the criterion for determining the reliability of the LOES analysis result, shows that increases as the $K_{aug}$ increases to 1.0, so there is limitation in verifying the reliability of the LOES analysis result. The MIL-STD-1797A recommends a numerical mismatch cost value of 10 to confirm the reliability of the analysis result, but the designer decides whether to accept the result of the analysis by referring to the bode plot trends of the higher order system (HOS) and LOES in the frequency band of interest.

### 7.3. Stability Margin Analysis

Table 3 shows the stability margin analysis results of the control system on the variation of the $K_{aug}$ and the synchronization of control surface feedback at Mach number 0.8 and altitude. The control surfaces such as asymmetric TEFs, asymmetric HTs and rudder were used for lateral-directional control of the aircraft.

**Table 3.** Result of stability margin analysis as a variation of additional augmentation gain and control surface synchronization in Mach 0.8, altitude 20 K, 1 g level flight.

| $K_{aug}$ | Surface Syn. | Asym. HT | | | Asym. TEF | | | Rudder | | | Spec. |
|---|---|---|---|---|---|---|---|---|---|---|---|
| | | Gain Margin (dB) | Phase Margin (°) | | Gain Margin (dB) | Phase Margin (°) | | Gain Margin (dB) | Phase Margin (°) | | |
| | | | Red | Inc | | Red | Inc | | Red | Inc | |
| 0.0 | on | 27.5 | N/A | N/A | 47.2 | N/A | N/A | 21.1 | −149.0 | 94.1 | satisfied |
| 0.6 | on | 18.5 | N/A | N/A | 17.0 | N/A | 149.5 | 13.7 | N/A | 59.0 | satisfied |
| 0.6 | off | 13.1 | N/A | N/A | 12.1 | N/A | 144.7 | 8.7 | N/A | 45.6 | satisfied |
| 0.8 | on | 17.0 | N/A | N/A | 15.0 | N/A | 91.1 | 12.4 | N/A | 49.8 | satisfied |
| 1.0 | on | 15.7 | N/A | 107.9 | 13.6 | N/A | 67.6 | 11.3 | N/A | 42.7 | unsatisfied |

The gain and phase margin of the model-based INDI control system, in which the synchronization filter is designed in the control surface feedback path and the $K_{aug}$ is set to 0.0, are sufficient to be 21.1 dB and 94.1° or more for rudder control surface. However, the stability margin of the control system significantly reduces as the $K_{aug}$ increases to 0.6. At this time, the gain margin is reduced by 9 dB from 27.5 dB to 18.5 dB for asymmetric HT, by 30.2 dB from 47.2 dB to 17 dB for asymmetric TEFs and by 7.4 dB from 21.1 dB to 13.7 dB for rudder control surfaces. In addition, the reduction in phase margin on the rudder control surface is 35.1°, which is quite large. Considering this stability margin analysis results, it can be seen that the hybrid INDI control that feedbacks the angular acceleration measured by the IMU significantly reduces the stability margin of the control system compared to the model-based INDI control. The gain margin is reduced by maximum 5 dB, and the phase margin is reduced by 13° when the $K_{aug}$ is set to 0.6 and the synchronization filter is not applied to the control surface feedback path. Considering this, the control surface feedback synchronization design for the time delay of the measured angular acceleration is also very important in the hybrid INDI control system. As already known, we can find that the hybrid INDI control system with the aircraft with statically unstable configurations has a significant reduction in stability margin depending on whether control surfaces feedback synchronization is applied [33,34].

As the value of the $K_{aug}$ increases, the stability margin of the control system decreases. When $K_{aug}$ is set to 1.0, the phase margin of the rudder control surface is 42.7°, that is, the control system does not satisfy any more the criteria above 45°. With reference to this analysis result, we select the maximum the $K_{aug}$ value as 0.8 and perform the sensitivity analysis and time-domain simulation to evaluate robustness and flying qualities.

### 7.4. Sensitivity Analysis for Various Uncertainties

The sensitivity analysis is performed to evaluate the robustness of the control system for the aerodynamic and control effectiveness uncertainties. The main coefficient factors select six aerodynamic for rolling and yaw moments for $\beta$, $p$ and $r$ and four control effectiveness for rolling and yaw moments for TEFs and rudder control surfaces, while the control effectiveness for the asymmetric HTs control surfaces is not considered as a robustness analysis item because it is not the main control surfaces of the lateral-directional axis maneuver. Figures 7 and 8 show the LOES and stability margin analysis results for model uncertainties.

The mismatch cost value tends to increase as the $K_{aug}$ increases, but this value is still less than 15, except for the uncertainties of −30% $L'_p$ and $L'_{\delta_{aa}}$ coefficients and +30% $N'_{\delta_r}$ coefficients. The model-based INDI control system is sensitive to model uncertainties, whereas the hybrid INDI control system is relatively robust. The variation of Dutch-roll mode frequency and damping ratio characteristics is reduced as the $K_{aug}$ increases. However, the roll time constant shows robust characteristics of the control system regardless of the $K_{aug}$ value except for the uncertainty of the $N'_\beta$ and $L'_p$ coefficients. From the perspective of the equivalent time delay characteristics, it has characteristics that decrease as the $K_{aug}$ increases, and the sensitivity of the control system to uncertainties is similar regardless of the value of $K_{aug}$.

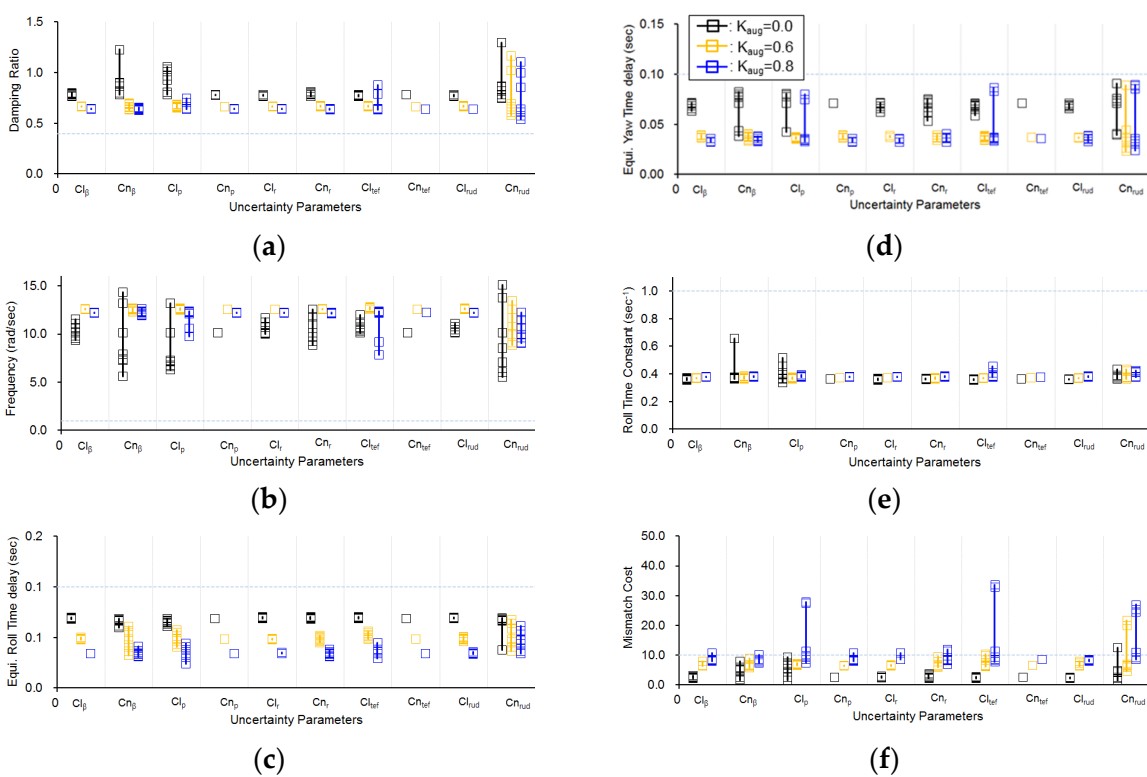

**Figure 7.** LOES analysis results of the control system on various uncertainties in M0.8, 20 Kft altitude (**a**) short-period damping, (**b**) short-period frequency, (**c**) equivalent roll time delay, (**d**) equivalent yaw time delay, (**e**) roll mode time constant, (**f**) LOES mismatch cost.

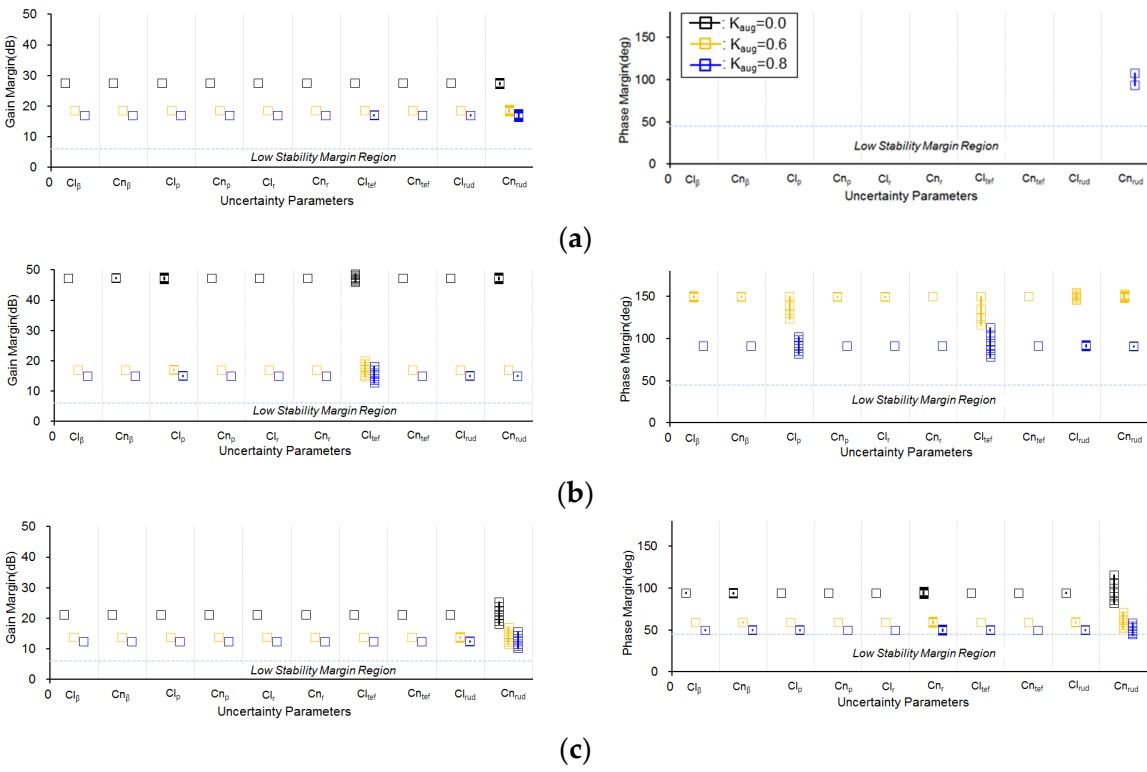

**Figure 8.** Stability margin analysis results of the control system on various uncertainties in M0.8, 20 Kft altitude (**a**) asymmetric HTs, (**b**) asymmetric flaperon, (**c**) rudder.

As already mentioned in the chapter on the stability margin analysis, the stability margin significantly decreases as the $K_{aug}$ increases. The phase margin of the control system is 49.7° in case that there are no uncertainties when the $K_{aug}$ is set to 0.8. In this condition, the phase margin of the control surface to the rudder control surface is reduced to 44.7° if there is an uncertainty of +30% of the control effectiveness of the rudder control surface, which is not satisfied with the phase margin criteria of 45° or more. In addition, the stability margin characteristics are shown to be sensitive to the uncertainties of the rolling moment for TEFs and the yawing moment for rudder control surface, while the control system is relatively robust, the characteristics to the $K_{aug}$ change and the sensitivity characteristics are mostly similar for the aerodynamic coefficient uncertainties.

Figures 9 and 10 show the results of time-domain simulations evaluating the lateral-directional response characteristics to the maximum roll and yaw doublet input with 30% uncertainties of the $L'_p$ and $N'_{\delta_r}$ coefficients. Here, (a) is the result of simulation with $K_{aug}$ set to 0.0, and (b) is $K_{aug}$ set to 0.8. The $L'_p$ uncertainty has a significant effect on the roll rate and roll attitude changes when $K_{aug}$ is set to 0.0. At this time, the maximum roll rate increases by 34°/s from 147°/s to 180°/s for −30% $L'_p$ uncertainties compared to when there is no uncertainty. Therefore, the roll attitude is increased by 23° from 83° to 106°. However, the amount of change in the roll rate response is considerably reduced to 13 °/s for $L'_p$ uncertainty in the hybrid INDI control. In addition, the $N'_{\delta_r}$ uncertainty affects the yaw rate and sideslip characteristics for maximum roll doublet input, which is quite robust when $K_{aug}$ is set to 0.8 compared to $K_{aug}$ is set to 0.0.

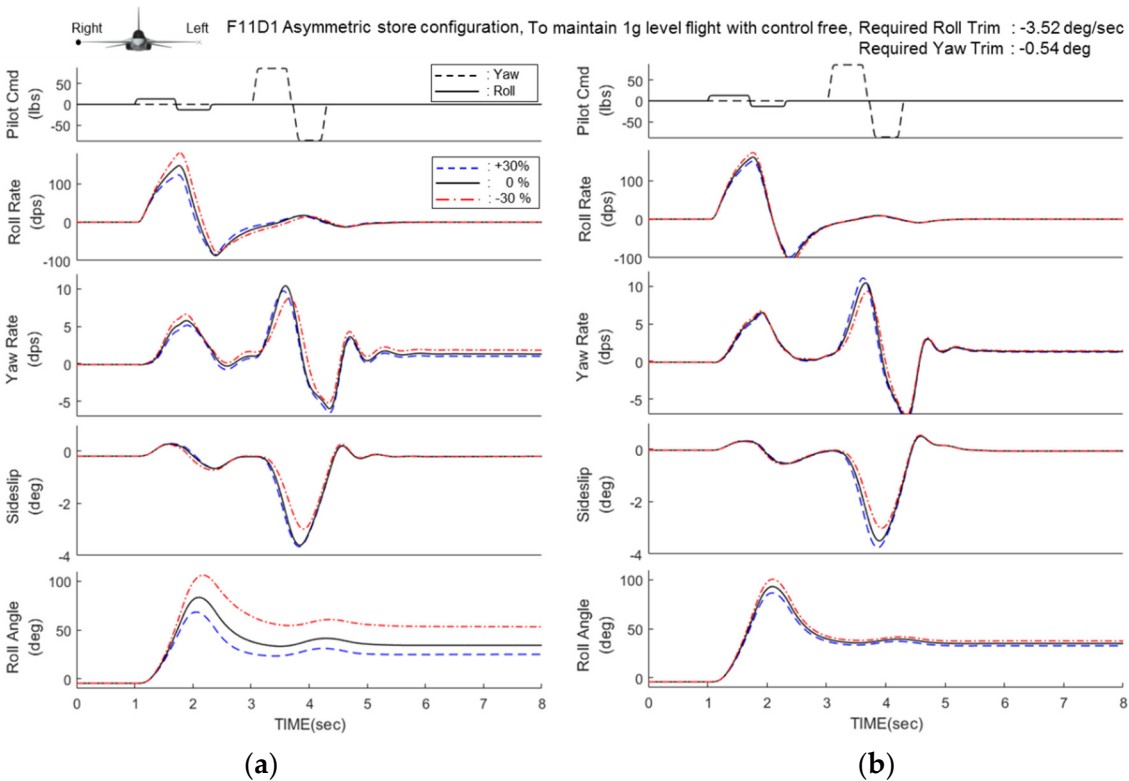

**Figure 9.** Robustness analysis results of the control system on $L'_p$ aerodynamic coefficient uncertainties ($\pm$30%) in M0.8, 20 Kft altitude (**a**) $K_{aug}$ = 0.0, (**b**) $K_{aug}$ = 0.8.

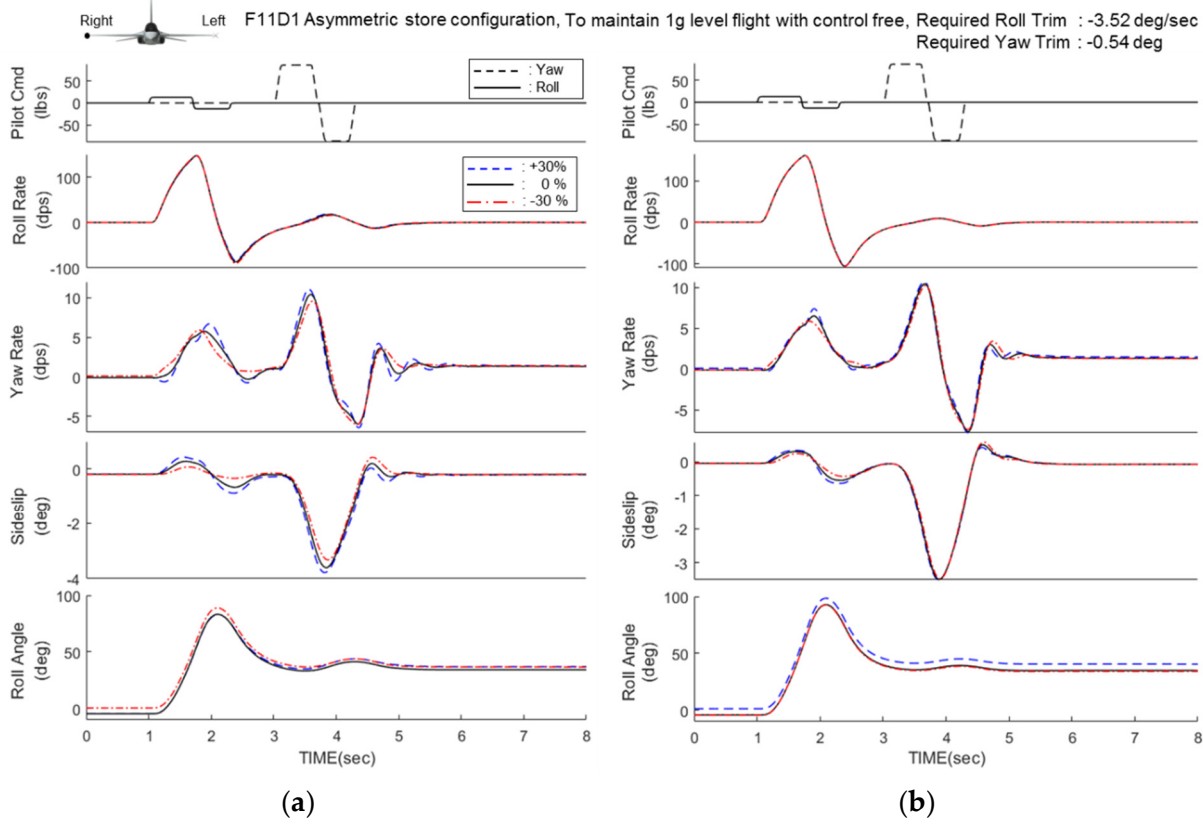

**Figure 10.** Robustness analysis results of the control system on $N'_{\delta_r}$ control effectiveness uncertainties ($\pm$30%) in M0.8, 20 Kft altitude (**a**) $K_{aug}$ = 0.0, (**b**) $K_{aug}$ = 0.8.

*7.5. Time-Domain Nonlinear Evaluation*

In this section, we present the results of evaluating the maximum roll performance, the roll transition response that occurs at store launch of the one wing and the unexpected roll response characteristic that occurs for pure longitudinal maneuvering in asymmetric store configurations as the $K_{aug}$ changes.

7.5.1. Transient for Store Launches

In the case of asymmetric launching of the stores, an abrupt transient response occurs due to ejection force and the movement of YCG changing by the mass distribution of both wings. This transient response adversely affects the ride qualities and degrades flying qualities of the aircraft. This section presents the simulation results of a transient response for the asymmetric store launches of the air-to-air store, the AIM-9.

Figure 11 shows the simulation result of launching the AIM-9 on the left wing in the symmetric store configuration. This store configuration is symmetric and does not require control stick and trim input to maintain 1 g level flight. The AIM-9 launch of the left wing moves the aircraft's CG to the right side due to mass distribution change the wings, resulting in a transient response of the right roll, and tilting the aircraft to the right. In model-based INDI control, with the $K_{aug}$ of 0.0, the roll transient response is quite large. At this time, the right roll with roll rate of 4°/s is generated by the AIM-9 launch of the left wing. The transient response of sideslip of −0.2 degrees and normal acceleration of 0.04 g occurs. Due to this effect, the roll attitude of the aircraft is 30° in 8 s after the AIM-9 launches. However, the transient response due to AIM-9 launch decreases proportionally as the $K_{aug}$ increases. The maximum transient response of the roll rate is 1.5°/s when the $K_{aug}$ is set to 0.8, decreasing by −3.5°/s compared to when the $K_{aug}$ value is 0.0. In addition, the roll attitude of the aircraft is significantly reduced from 30° to 6° in 8 s after the AIM-9 launches.

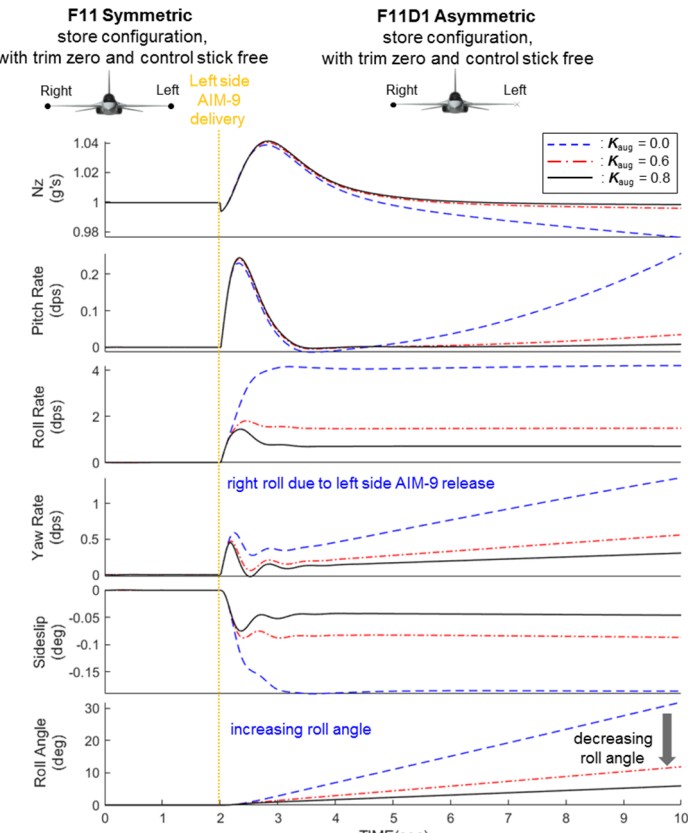

**Figure 11.** Transient response to asymmetric weapon delivery in M0.8, 20 Kft altitude.

### 7.5.2. Unexpected Roll Motion for Longitudinal Maneuver

Generally, the pilot uses a manual trim system for the aircraft to maintain 1 g level flight in an asymmetrical store configuration. In the state of maintaining level flight in the specific flight condition, the change of AoA and airspeed caused by pure longitudinal maneuver affects the lateral-directional dynamic characteristics of the aircraft by the mass unbalance and lift disturbance of both wings, which affects the amount of trim for level flight. Therefore, the pilot continuously uses the roll control stick input for wing level for pure longitudinal maneuver, and this increases the pilot's workload for performing the mission tasks.

Figure 12 shows the results of evaluating the response of the aircraft at the pull-up maneuver after maintaining 1 g level flight using the manual trim in the asymmetric store configuration. For 1 s after the control stick input, the AoA does less change than the trim AoA of 1.6°. Therefore, the aircraft can maintain level flight without lateral-directional motion. In case that the $K_{aug}$ value is set to 0.0, the airspeed decreases, and the AoA increases after 2 s from pull up maneuver. For this reason, the aircraft cannot maintain the level flight and unexpected rolling motion in the left direction occurs. The maximum roll rate is $-10.2°/s$, the yaw rate is $-2°/s$, the sideslip is $0.8°$, and the roll attitude of the aircraft is $-57.9°$ around 10 s. This phenomenon appears to be that the trim value used to maintain 1 g level flight of the aircraft at the initial 1.5° AoA does not match the initial trim value due to the effect of mass unbalance and lift distribution of both wings as the AoA increases. In other words, this is because the amount of trims needed for maintaining the aircraft in level flight of the asymmetric store configuration varies depending on the airspeed and AoA. Due to this characteristic, the pilot should continuously use the roll control stick to maintain level flight for pure longitudinal maneuvering, increasing the pilot's workload and degrading the efficiency of combat missions. In the hybrid INDI control, the unexpected roll response that occurs as the AoA increases significantly decreases as the $K_{aug}$ is increased. The roll rate is $-4.5°/s$, the yaw rate is $-0.8°/s$, the sideslip is decreased to $0.6°$ and the roll attitude is $-20.1°$ reduced by more than 65% when the $K_{aug}$ is set to 0.6 compared to the model-based INDI control. In addition, the roll attitude is $-9.78°$, and it decreases to 83%.

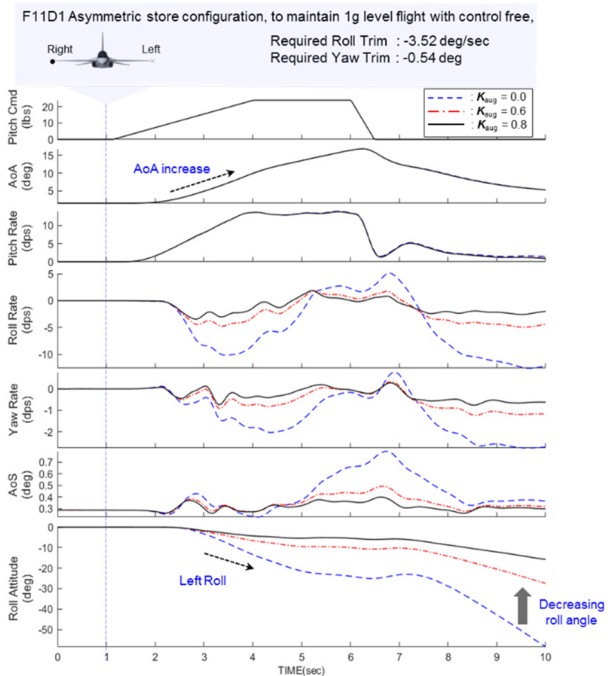

**Figure 12.** Simulation results of lateral-directional flying qualities to pull up maneuver in M0.8, 20 Kft altitude.

Considering this simulation result, the hybrid INDI control which partly uses the measured angular acceleration is effective control methods that improve the pilot's ride qualities and flying qualities by effectively reducing the transient response when asymmetric store launches. In addition, this control method significantly reduces the unexpected roll motion that occurs for pure longitudinal maneuver in an asymmetric store configuration and improves flying qualities in combat missions by reducing the pilot's workload.

### 7.5.3. Maximum Roll Performance

The simple roll control method proposed by C. Kim et al. [35] that increases the control gain of the roll rate feedback path can improve in preventing the unexpected roll response that occurs during pure longitudinal maneuvering in an asymmetric store configuration. However, this control method has the problem of increasing the structural load of the aircraft by rapidly increasing the deflection of the TEF control surface due to the control gain increased in the maximum 360° roll maneuver. In consideration of this, C. Kim et al. proposed a blend roll control structure that increases the control gain of the roll rate feedback path within a limited range of roll rate and roll control stick force. However, this control method has the problem of increasing design complexity.

This section presents the result of evaluating the control surface deflection trend by performing the maximum 360° roll maneuver to check the structural load influence in the hybrid INDI control. Figure 13 shows the 6-DOF nonlinear simulation results of a maximum 360° left roll maneuver with a variation of $K_{aug}$ in Mach number 0.8 and altitude 20 Kft 1 g level flight. The initial roll performance slightly improves as a deflection of the TEF control surface slightly increases with respect to the roll control stick input as the $K_{aug}$ value increases, while the maximum roll performance is −190°/s regardless of the $K_{aug}$ value. Therefore, this control method does not affect the increase of the structural load as it can achieve the maximum roll performance of the aircraft without significant difference of the control surface deflection. The AoS decreases from 1.0° to 0.8°, and the yaw rate increases from 1.8°/s to 3.4°/s as the $K_{aug}$ increases. The maximum normal acceleration increases by 0.14 g from 1.2 g to 1.34 g in the positive direction, and the pitch rate increases by 1.61°/s from 1.69°/s to 3.3°/s. However, the magnitude of the pitch and yaw axes coupling response to the maximum roll maneuver is relatively small, so it does not affect flying qualities.

Considering the evaluation result, the hybrid INDI control method provides consistent roll performance to the aircraft without directly affecting the aircraft's achieving maximum roll performance as a variation of the $K_{aug}$, and it has little effect on the structural load since the amount in deflection difference is small although the increase of the $K_{aug}$ value affects the initial TEF deflection.

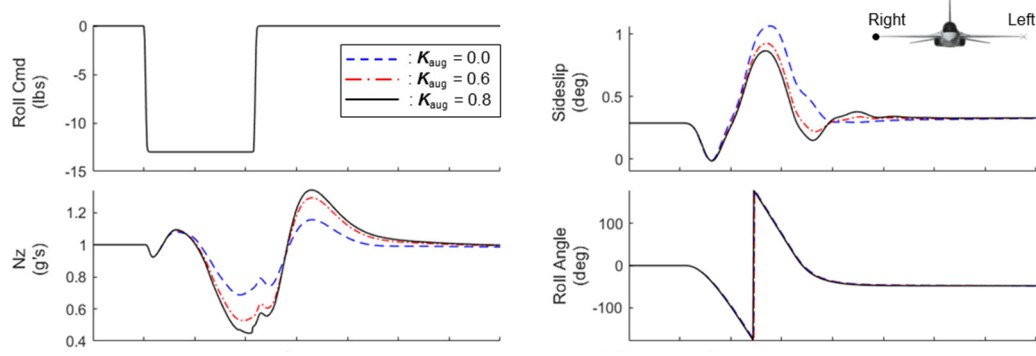

**Figure 13.** *Cont.*

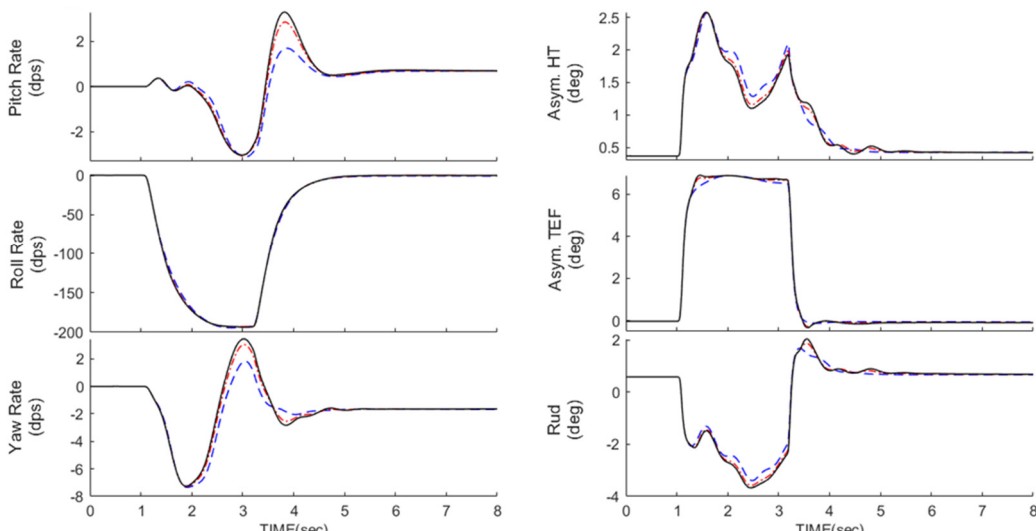

**Figure 13.** Simulation result of maximum roll rate performance characteristics to max 360 degree roll maneuver in M0.8, 20 Kft altitude.

## 8. Conclusions

Modern highly maneuverable fighters are equipped with various stores. When stores mounted on one side wing are launched for air-to-air combat missions, abrupt roll motion is generated due to ejection force and movement of lateral center-of-gravity by changing the mass distribution of both wings. This transient response not only adversely affects the pilot's ride comfort but also degrades flying qualities. In asymmetric store configuration after store launches, the pilot should use a manual trim system to maintain 1 g level flight and the required amount of trim varies according to the AoA, altitude and airspeed conditions. For this reason, even if the 1 g level flight is maintained using the manual trim system in a specific altitude and speed, the change in the AoA and airspeed that occurs affects the imbalance of the lift force distribution on both wings, resulting in unexpected roll motion in pure longitudinal maneuvering. In other words, it means that the pilot should continuously use the roll control stick input to achieve a pure longitudinal axis maneuver, which significantly increases the workload. This leads to poor flying qualities in asymmetric store configurations.

Besides, modern fighters have a lateral asymmetric characteristic in lateral axis due to the installation of infra-red search and track (IRST) and electronic optics targeting pod (EOTGP). To overcome this problem, the fighter aircrafts such as JAS-39 and EF-2000 adopt an automatic trim system on three axes. However, this design increases the design complexity and increases the computational throughput of the flight control computer because the integrator should be used in the lateral-directional inner-loop control architecture.

In this study, we proposed the Hybrid INDI control that combines sensor- and model-based INDI control to improve this problem. This control method effectively reduces the transition response of the roll axis for store launch and significantly improves flying qualities by reducing unexpected roll motion for pure longitudinal maneuvering in asymmetric store configuration. In addition, it is possible to secure the reliability of the control algorithm for flight safety by using model-based INDI control applied to production aircrafts and fairly robust control method against various uncertainties of major aerodynamic and control effectiveness properties compared to other classical control methods.

## 9. Future Works

The proposed control method still has the disadvantage of decreasing the stability margin of the control system compared to the model-based INDI control. In the future, we plan to research a new control algorithm that further increases the stability margin and reduces phase lag based on the hybrid INDI control.

**Author Contributions:** Conceptualization, C.-h.J., C.-s.K., and B.-S.K.; methodology, C.-h.J., and C.-s.K.; software, C.-s.K.; validation, C.-h.J., and C.-s.K.; formal analysis, C.-s.K.; investigation, C.-h.J., and C.-s.K.; resources, C.-h.J., and C.-s.K.; data curation, C.-s.K.; writing—original draft preparation, C.-h.J., C.-s.K.; writing—review and editing, C.-h.J., C.-s.K.; visualization, C.-h.J., C.-s.K.; supervision, B.-S.K. All authors have read and agreed to the published version of the manuscript.

**Funding:** This research received no specific grant from any funding agency in the public, commercial or not-for-profit sectors.

**Institutional Review Board Statement:** Not applicable.

**Informed Consent Statement:** Not applicable.

**Data Availability Statement:** The data used to support the findings of this study are available from the corresponding author upon request.

**Acknowledgments:** The authors would like to deliver their sincere thanks to the editors and anonymous reviewers.

**Conflicts of Interest:** The authors declare that there are no conflicts of interest regarding the publication of this paper.

**Nomenclature**

The following nomenclatures are used in this manuscript:

| | |
|---|---|
| $x$ | state vector |
| $u$ | control input vector |
| $f$ | nonlinear state dynamic function |
| $g$ | nonlinear control distribution function |
| $\Delta u$ | incremental control command ($°$) |
| $\Delta d$ | virtual control command ($°$) |
| $u_0$ | previous control command ($°$) |
| $K_{aug}$ | additional augmentation control gain |
| $f_{obm}$ | nonlinear state dynamic function of OBM |
| $g_{obm}$ | nonlinear control distribution function of OBM |
| $\dot{x}_{des}$ | rate of desired state vector ($°/\mathrm{s}^2$) |
| $\dot{x}_{obm}$ | rate of state vector calculated from OBM ($°/\mathrm{s}^2$) |
| $\dot{x}_{add}$ | rate of state vector of additional augmentation control sensor ($°/\mathrm{s}^2$) |
| $K_{r1}$ | flying quality parameter of roll command |
| $K_{r2}$ | flying quality parameter of roll rate feedback |
| $K_{y1}$ | flying quality parameter of yaw command |
| $K_{y2}$ | flying quality parameter of sideslip feedback |
| $K_{y3}$ | flying quality parameter of sideslip rate feedback |
| $p_s$ | stability axis roll rate ($°/\mathrm{s}$) |
| $p_{s,cmd}$ | stability axis roll rate command ($°/\mathrm{s}$) |
| $\tau_{roll}$ | roll time constant (s) |
| $\beta$ | angle of sideslip ($°$) |
| $\beta_{cmd}$ | angle of sideslip command ($°/\mathrm{s}$) |
| $\omega_{dr}$ | dutch-roll frequency (rad) |
| $\zeta_{dr}$ | dutch-roll damping ratio |
| $\dot{p}$ | roll angular acceleration ($°/\mathrm{s}^2$) |
| $\dot{r}$ | yaw angular acceleration ($°/\mathrm{s}^2$) |
| $\dot{p}_{des}$ | desired roll angular acceleration ($°/\mathrm{s}^2$) |
| $\dot{r}_{des}$ | desired yaw angular acceleration ($°/\mathrm{s}^2$) |
| $p$ | roll rate ($°/\mathrm{s}$) |
| $q$ | pitch rate ($°/\mathrm{s}$) |
| $r$ | yaw rate ($°/\mathrm{s}$) |
| $I_{ii}$ | principal moment of inertia (slug-ft$^2$) ($i = x, y, z$) |
| $I_{ij}$ | production moment of inertia (slug-ft$^2$) ($i = x, y, z, j = x, y, z$) |
| $L$ | rolling moment of the aircraft |
| $N$ | yawing moment of the aircraft |

| | |
|---|---|
| $L'_k$ | linearized rolling moment for $k$ ($k = \beta, p, r, \delta_{ea}, \delta_r$) |
| $N'_k$ | linearized yawing moment for $k$ ($k = \beta, p, r, \delta_{ea}, \delta_r$) |
| $\delta_k$ | control surface deflection for $k$ ($k = ea, aa, r$) |
| $\boldsymbol{K_{aug}}$ | additional augmentation control gains |
| $G^u_{ac}$ | aircraft plant dynamics for control surface |
| $\dot{x}_{fb}$ | total angular acceleration feedback ($^\circ/s^2$) |
| $H_{sync}$ | synchronization filter matrix |
| $H^{total}_2$ | angular acceleration sensor model with aircraft dynamics |
| $H^{total}_1$ | AoA and IMU sensor models with aircraft dynamics |
| $\zeta_{syn}$ | damping ratio of 2nd order synchronization filter (rad) |
| $\omega_{syn}$ | natural frequency of 2nd order synchronization filter |

**Abbreviations**

The following abbreviations are used in this manuscript:

| | |
|---|---|
| AoA | angle of attack |
| AoS | angle-of-sideslip |
| CA | control allocation |
| EOTGP | electronic optics targeting pod |
| DLR | german aerospace center |
| HARV | high angle-of-attack research |
| HOS | high-order system |
| HT | horizontal tail |
| IRST | infra-red search and track |
| IMU | inertial measurement unit |
| INDI | incremental nonlinear dynamic inversion |
| JSF | joint strike fighter |
| LOES | low-order equivalent system |
| NASA | national aeronautics and space administration |
| NLR | netherlands aerospace centre |
| OBM | on-board model |
| RESTORE | reconfigurable control for tailless aircraft |
| RSRI | rolling surface-to-rudder interconnect |
| STOVL | short take-off/vertical landing |
| TEF | trailing edge flap |
| VAAC | vectored thrust aircraft advanced control |

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
