# Peer review of "A Hybrid Incremental Nonlinear Dynamic Inversion Control for Improving Flying Qualities of Asymmetric Store Configuration Aircraft"

_aerospace, doi:10.3390/aerospace8050126_

Round 1

Reviewer 1 Report

This manuscript provides quite practical technique for highly maneuverabilty fighter aircraft. The motivation and presentation are quite reasonable and clearly written. It is believed that this article is worthy of being published to readers.

Recommendation:

The abbreviation in the title is not recommended. INDI should be replaced to Incremental Nonlinear Dynamic Inversion. 

p4, l158, " For conventional uses where small perturbations form trim conditions, the function ? is linear in u". This reviewer believes that the  linearized model does not provide any relationship with the other part of the manuscript. This sentence seems make unnecessary misunderstanding. It could be removed.

Author Response

We appreciate the Reviewer's work on our paper. The comment provided was useful in revising the manuscript and improve the shortcomings of the manuscript. Thank you so much.

We answered the response to the reviews in the attached files.

Reviewer 2 Report

The work is interesting.
I recommend small changes in the attached file.

Author Response

(The authors gave the same response as above.)
